# Rational combination of oncolytic vaccinia virus and PD-L1 blockade works synergistically to enhance therapeutic efficacy

Zuqiang Liu[1,2], Roshni Ravindranathan[1,2], Pawel Kalinski[1,2,3], Z. Sheng Guo[1,2] & David L. Bartlett[1,2]

Both anti-PD1/PD-L1 therapy and oncolytic virotherapy have demonstrated promise, yet have exhibited efficacy in only a small fraction of cancer patients. Here we hypothesized that an oncolytic poxvirus would attract T cells into the tumour, and induce PD-L1 expression in cancer and immune cells, leading to more susceptible targets for anti-PD-L1 immunotherapy. Our results demonstrate in colon and ovarian cancer models that an oncolytic vaccinia virus attracts effector T cells and induces PD-L1 expression on both cancer and immune cells in the tumour. The dual therapy reduces PD-L1$^+$ cells and facilitates non-redundant tumour infiltration of effector CD8$^+$, CD4$^+$ T cells, with increased IFN-$\gamma$, ICOS, granzyme B and perforin expression. Furthermore, the treatment reduces the virus-induced PD-L1$^+$ DC, MDSC, TAM and T$reg$, as well as co-inhibitory molecules-double-positive, severely exhausted PD-1$^+$ CD8$^+$ T cells, leading to reduced tumour burden and improved survival. This combinatorial therapy may be applicable to a much wider population of cancer patients.

[1] The University of Pittsburgh Cancer Institute, University of Pittsburgh School of Medicine, Pittsburgh, Pennsylvania 15213, USA. [2] Department of Surgery, University of Pittsburgh School of Medicine, Pittsburgh, Pennsylvania 15213, USA. [3] Department of Immunology, University of Pittsburgh School of Medicine, Pittsburgh, Pennsylvania 15213, USA. Correspondence and requests for materials should be addressed to Z.S.G. (email: guozs@upmc.edu) or to D.L.B. (email: Bartlettdl@upmc.edu).

Oncolytic viruses (OVs) are native or recombinant viruses which can selectively kill cancer cells and associated stromal cells directly by oncolysis, indirectly by immune mediated clearance of cancer cells, or targeting of tumour vasculature. Importantly, a systemic effect can be mediated through the induction of systemic anti-tumour immunity, especially when OVs are armed with immunostimulatory genes such as GM-CSF (refs 1,2). Talimogene laherparepvec (T-VEC, Imlygic), a herpes simplex virus expressing GM-CSF, was associated with improved overall survival after a local, intratumoral injection of the virus, and is the first OV-based drug approved by the Food and Drug Administration[3].

Cancer immunotherapy has joined the ranks of surgery, radiation and chemotherapy as a tool for cancer treatment[4]. It consists of active immunotherapy such as cancer vaccines[5], passive immunotherapy such as adoptive cell transfer, including ex vivo expanded tumour-infiltrating lymphocyte and CAR-T cells[6,7], and strategies to modulate the immunosuppressive tumour microenvironment (TME) such as using antibodies that bind to and modulate the function of immune checkpoints (such as CTLA-4 and PD-1/PD-L1) (refs 8,9). Oncolytic virotherapy has been classified as another form of novel immunotherapy[1,2,10], and in addition to herpes virus, vectors such as vaccinia virus have demonstrated promise in this arena. Vaccinia virus is highly immunogenic and has properties that make it an ideal oncolytic immunotherapy vector[11]. Preclinical murine studies demonstrate significant anti-tumour efficacy and systemic anti-tumour immunity, using a tumour-selective oncolytic vaccinia virus expressing immunogenic transgenes[12–15]. An oncolytic vaccinia virus armed with GM-CSF (Pexa-Vec) was associated with a 15% objective response rate in patients with advanced hepatocellular carcinoma in a randomized phase II clinical trial[15]. We have shown recently that a tumour-selective Western Reserve strain oncolytic vaccinia virus, vvDD (without any immunogenic transgene), was safe and exhibited some anti-tumour effects in patients with advanced solid tumours in phase I clinical trials[16,17]. However, overall the therapeutic efficacy in patients has been limited, especially when the tumour is poorly immunogenic, and the TME highly immunosuppressive. We have recently demonstrated in a poorly immunogenic (MC38 colon) tumour model, however, that an oncolytic vaccinia virus expressing the T-cell attracting chemokine, CXCL11, can attract effector cells into the TME and induce specific systemic anti-tumour immunity[18].

In the last few years, one particularly exciting area of immunotherapy has been the use of anti-PD1/PD-L1 antibodies for cancer treatment. Chen and co-workers showed that tumour-associated PD-L1 (or called B7-H1) promotes T-cell apoptosis which could be a potential mechanism of immune evasion[19]. Earlier it had been shown that engagement of PD-1 on lymphocytes by a novel B7 family member (later found to be PD-L1) leads to negative regulation of lymphocyte activation[20]. It was also reported that the expression of PD-1 is upregulated on exhausted CD8$^+$ T cells from mice chronically infected with lymphocytic choriomeningitis virus, and PD-1/PD-L1 blockade enhanced virus-specific CD8$^+$ T-cell responses and reduced viral load[21]. The PD-1/PD-L1 mediated immune escape has subsequently been confirmed during HIV, HBV and HCV infections[22]. It was recognized that tumour cells express PD-L1 on their surface, inactivating immune effector cells. Those tumours with high levels of PD-L1 on their surface and a lymphocytic infiltrate have been shown to respond well to anti-PD-1/anti-PD-L1 therapy, including melanoma, Hodgkin's lymphoma, non-small-cell lung, bladder, gastric, renal and ovarian cancers[23]. In this regard, it is interesting to note the hints that virus-associated cancers respond at high rates to

PD1 pathway blockade[8]. Most likely, this is due to the fact that oncogenic viruses often induce chronic inflammation and secret cytokines such as IFN-γ that induce PD-1/PD-L1 expression[24–27]. Nevertheless, most cancers do not associate with viruses. This anti-PD1/PD-L1 therapy does not work well in most cancer types where there are minimal lymphocytic infiltrates, and very low expression levels of PD-L1 (ref. 28). Even in the applicable types of cancer such as melanoma, this approach is effective in only about 15–25% of the patients. Therefore, expanding the successful application of this treatment would be significant.

We hypothesized that OV will upregulate PD-L1 in the TME as a means of self-protection, and that in tumours with low immunogenicity and minimal PD-L1 expression, a vaccinia virus expressing CXCL11 (vvDD-CXCL11 or called VV) will enhance T-cell infiltration into the tumour and upregulate the expression of PD-L1. Combined treatment with anti-PD-L1 will then lead to effective tumour clearance. Our study tests a rational combination therapy of oncolytic vaccinia and PD-L1 blockade in animal tumour models, with the potential for improving immunotherapy in cancer patients, including those who have naturally low/no PD-L1-expressing tumours.

## Results

**PD-L1 upregulation in cancer cells and in tumour.** We initially asked if our commonly used murine cancer cell lines naturally express PD-L1, and if the infection of oncolytic vaccinia virus would impact PD-L1 expression on tumour cells. MC38-luc and ID-8-luc cancer cells were either mock-infected or infected with vvDD, and harvested 24 h later. PD-L1 expression on the cells was measured by both flow cytometry and RT–qPCR assays. The basal levels of PD-L1 expression were very low in ID8-luc and a bit higher in MC38-luc cancer cells. Upon infection, PD-L1 expression was enhanced in both cell lines as measured by flow cytometry and by qPCR (Fig. 1a,b). We also performed qPCR assays on a panel of human cancer cell lines, representing cancers from brain, breast, colorectal, lung, ovarian and pancreas. In all cases, the PD-L1 mRNA expression in the cancer cells was enhanced post vvDD infection, ranging from 2 to 16-fold (Fig. 1c). This is also the case in various other murine cancer cell lines (Supplementary Fig. 1). In summary, these data demonstrate that PD-L1 expression is upregulated in cancer cells post vaccinia virus infection in vitro.

To explore how vaccinia virus infection impacts PD-L1 expression in vivo in the TME, MC38 colon tumour-bearing C57BL/6 (B6) mice were injected with vvDD or PBS intratumorally, and tumour tissues were collected 4 days later. PD-L1 expression was significantly increased in virus-treated tumour tissues compared with PBS-treated ones when analysed by RT–qPCR (Fig. 2a). To further dissect the cellular origins of increased PD-L1, we analysed PD-L1 expression on tumour and other non-immune cells (CD45$^-$), MDSCs (CD45$^+$CD11b$^+$Gr1$^+$) and macrophages (CD45$^+$CD11b$^+$F4/80$^+$) in the TME by flow cytometry. A significant increase in the PD-L1 expression on tumour cells was observed via both the absolute number and percentage of PD-L1$^+$ CD45$^-$ cells (Fig. 2b, Supplementary Fig. 2a). A slight increase in the percentage of PD-L1 expression on MDSC and TAM was also observed (Supplementary Fig. 2b,d); and the absolute number of PD-L1$^+$ MDSC and PD-L1$^+$ TAM was significantly increased as the overall numbers of MDSC and TAM were elevated post virus infection (Fig. 2c,d, Supplementary Fig. 2c,e).

**Combination synergy in anti-tumour effects.** To test our hypothesis that OV and anti-PD-L1 therapies might work in synergy, we studied intraperitoneal (i.p.) MC38-luc colon and

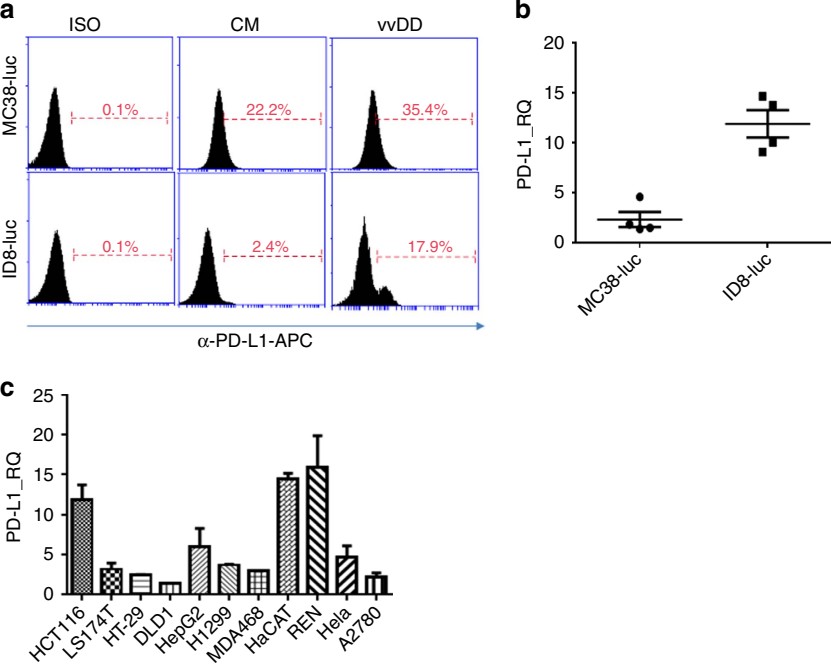

**Figure 1 | PD-L1 is elevated post vvDD infection *in vitro*.** (**a**) MC38-luc colon cancer or ID8-luc ovarian cancer cells ($4 \times 10^5$ cells each) were mock-infected or infected with vvDD. These cells were harvested 24 h later, blocked with α-CD16/32 Ab and then stained with α-PD-L1 for flow cytometry. (**b**) Total RNA was extracted from the harvested tumour cells and used in RT–qPCR to determine PD-L1 expression. In **a**, ISO: isotype IgG control used for staining; CM: condition medium. Data are presented as individuals, mean $+/-$ s.d. (**c**) Cells from a panel of human cancer cell lines representing colorectal, ovarian, lung, cervical cancer and mesothelioma were infected or mock-infected with vvDD for 24 h and total RNA was prepared and subjected to RT–qPCR to determine the relative expression of PD-L1. The values on the *y* axis indicate the ratio of PD-L1 expression between infected versus mock-infected cancer cells. Data are presented as mean $+/-$ s.d.

ID8-luc ovarian tumour models in B6 mice. Because vvDD-CXCL11 has previously been demonstrated to be superior to vvDD by attracting an increased number of effector T cells[14], we have used this virus (short-named as VV in the figures) in the remaining experiments. It should be noted, however, that direct comparisons between vvDD and VV when combined with α-PD-L1 Ab did demonstrate only a trend of better, yet not statistically significant difference in therapeutic efficacy for the CXCL11 virus (data not shown). The tumour-bearing mice were treated with PBS, α-PD-L1 Ab, VV or VV plus α-PD-L1 Ab. In order to monitor tumour growth, the mice were killed at days 2, 5 and 13 post first treatment (days 7, 10, 18 post tumour inoculation), and tumour nodules were collected and weighed. Two days after treatment, VV plus α-PD-L1 or VV alone treated mice had smaller tumour burden than those in PBS-treated or α-PD-L1-treated mice (Fig. 3a–d). By days 5 and 13, either α-PD-L1 or VV-treated mice had similarly reduced tumour burden, but the dual treatment led to a significantly reduced tumour burden compared to monotherapies (Fig. 3c,d).

We also used live animal bioluminescence imaging to monitor the kinetics of tumour growth in mice. Both α-PD-L1 antibody alone and VV treatment caused delays in tumour growth; however, the dual therapy resulted in the most tumour inhibition, statistically improved over either monotherapy on day 9 (Fig. 4a,b). In terms of animal survival, both VV treatment and PD-L1 blockade led to improvements in survival, but the combination of VV plus α-PD-L1 Ab led to the best overall survival in both colon cancer (Fig. 4c) and ovarian cancer (Fig. 4d). We also asked whether different time points for the reagent administration could impact the therapeutic effect elicited by the combination therapy. We fixed the first treatment time at day 5 post tumour cell injection and applied treatment using

different timing schemes: (1) α-PD-L1 Ab followed in 2 days by VV treatment; (2) VV treatment followed in 2 days by α-PD-L1 Ab therapy; and (3) simultaneous administration of both α-PD-L1 Ab and VV. The survival data showed that the simultaneous administration achieved the best therapeutic effect (Supplementary Fig. 3).

**The α-PD-L1 treatment reduced PD-L1[+] cells in the TME.** To explore the mechanisms of combination treatment-elicited anti-tumour effects, we first investigated the PD-L1 expression on a variety of types of stromal cells as well as cancer cells in tumour tissues. At day 2 or 5 post first treatment, VV infection led to increased PD-L1 expression in CD45[−] cells in both the percentage and absolute number (Fig. 5a,b, Supplementary Fig. 4a,b). Treatment with α-PD-L1 Ab (either α-PD-L1 Ab alone or VV plus α-PD-L1 Ab) led to reduced PD-L1—lower than the control group (PBS).

We have also investigated the effects on MDSCs and TAMs (Supplementary Fig. 5c–f). First, we observed an increase in both G-MDSC and M-MDSC in the CD45[+] cell population after dual treatment (Supplementary Fig. 4). Indeed, the virus treatment alone enhanced the number of G-MDSC in the tumour tissue (Supplementary Fig. 4a), confirming the previous finding by Yang and associates[29]. Then, we analysed PD-L1[+] MDSCs and TAMs in greater detail. Their absolute numbers, percentage and intensity of expression (MFI) in the tumours from mice treated with either α-PD-L1 Ab or VV plus α-PD-L1 Ab were smaller when compared with those treated with VV alone (Fig. 5c). This reduction happened to both G-MDSCs and M-MDSCs (Fig. 5d,e). The patterns for infiltrated TAM were quite different. Treatment with VV led to an increase of

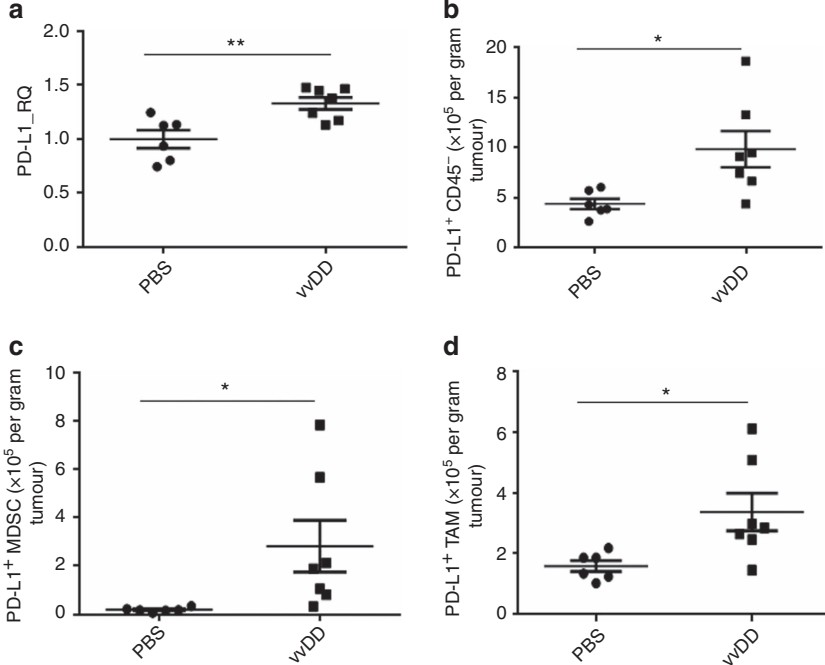

**Figure 2 | PD-L1 is elevated in cancer tissue post vvDD treatment *in vivo*.** B6 mice were subcutaneously inoculated with MC38-luc cells ($4 \times 10^5$ per mouse). PBS or vvDD was intratumorally injected at $2 \times 10^8$ pfu per tumour when the s.c. tumour area reached $5 \times 5$ mm$^2$. Tumour tissues were collected from PBS or virus-treated mice 4 days post treatment. Collected tumour tissues were weighed and incubated in RPMI 1640 medium containing 2% FBS, 1 mg ml$^{-1}$ collagenase, 0.1 mg hyaluronidase, and 200 U DNase I at 37 °C for 1–2 h to make single cell preparations. The single cells were used for extraction of total RNA for RT–qPCR assay (**a**), or they were blocked with α-CD16/32 Ab and then stained with antibodies against CD45, CD11b, Gr1, PD-L1, F4/80 to determine PD-L1 expression on tumour cell (CD45$^-$) (**b**), MDSC (CD45$^+$CD11b$^+$Gr1$^+$) (**c**) and TAM (CD45$^+$CD11b$^+$F4/80$^+$) (**d**). In this experiment, the anti-PD-L1 Ab clone 10F.9G2 was used for analysis. Significant differences are indicated by *($P < 0.05$) or ** ($P < 0.01$) determined by $t$-test. In this and other figures, the standard symbols for $P$ values are *$P < 0.05$; **$P < 0.01$.

PD-L1$^+$TAM1 cells in both percentage and MFI (Fig. 5f), while α-PD-L1 Ab led to greatly reduced PD-L1$^+$TAM1 cells. The dual treatment neutralized each other, leading to levels comparable to the PBS-treated group. The same patterns of changes also happened to PD-L1$^+$TAM2 (Fig. 5g). These data demonstrate that α-PD-L1 treatment reduces PD-L1 in the subclasses of MDSC and TAM cells, including G-MDSC, M-MDSC, TAM1 and TAM2 cells in the TME.

We have also examined the changes of DC and NK cells. The treatment of VV did not have any impact on the level of PD-L1$^+$DC, yet the treatment with anti-PD-L1 Ab, alone or in combination with VV, almost completely knocked out the PD-L1$^+$ DC (Fig. 5c). No changes were observed with the level of NK cells, even though there was a trend of increasing number of NK1.1$^+$ NK cells in the group treated with anti-PD-L1 Ab (Supplementary Fig. 5g).

**Dual therapy enhances beneficial repertoire of anti-tumour T cells.** For CD8$^+$ T cells, the percentage (in CD45$^+$ cells) was not enhanced in the dual treatment group when compared to any singular treatment group on day 2 (Supplementary Fig. 6a), but significantly elevated by day 5 (Fig. 6a). Two activation markers for CD8$+$ T cells, IFN-γ and ICOS were increased significantly in the group treated with both VV and α-PD-L1 antibody (Fig. 6b–d). We then examined four co-inhibitory molecules expressed on PD-1$^+$CD8$^+$ T cells. For LAG-3 expression, α-PD-L1 Ab led to a trend towards a decrease ($P = $ ns), VV treatment led to a significant reduction, and the dual treatment led to further significant reduction (Fig. 6e. $P < 0.05$ for VV versus PBS;

$P < 0.01$ between PBS and VV + α-PD-L1). As for TIM3 or CTLA-4 expression, α-PD-L1 did not have any effect, but VV or VV + α-PD-L1 led to almost complete eradication of the double-positive cells (Fig. 6f,g; between PBS and dual treatments, $P < 0.01$ for TIM-3; $P < 0.05$ for CTLA4). For TIGIT expression on PD-1$^+$CD8$^+$ T cells, neither α-PD-L1 nor VV had much of an impact, but the dual treatment led to a significant reduction (Fig. 6h; $P < 0.01$). In light of conclusion from previous studies that these double co-inhibitory molecules-positive cells are more severely exhausted T cells[30], our novel findings have significant implications for basic immunology as well as novel therapeutic strategies involving immune checkpoint blockade and/or OV.

We also examined subclasses of CD4$^+$ T cells including the regulatory T cells (Treg) in the TME. The ratio of Treg over the total CD4$^+$ T-cell population did not change with VV treatment, but reduced significantly in the groups treated with either α-PD-L1 or VV + α-PD-L1 (Fig. 6i). As a result, the ratio of CD8$^+$ T cells over Treg cells increased in the groups treated with VV or VV + α-PD-L1 (Fig. 6j; $P < 0.05$ between PBS and VV; $P < 0.01$ between VV and VV + α-PD-L1). Both the ratio of IFN-γ$^+$FoxP3$^-$CD4$^+$ T cells over FoxP3$^-$CD4$^+$ T cells and the absolute numbers of those activated IFN-γ$^+$FoxP3$^-$CD4$^+$T cells increased in the dually treated group on day 5 (Fig. 6k,i). In addition, the absolute number of FoxP3$^-$CD4$^+$ T cells among CD45$^+$ cells also increased on day 5 (Supplementary Fig. 6e).

The time course of the ratio of CD8$^+$/Treg was analysed in more detail. On day 2, this ratio was slightly elevated in α-PD-L1 and VV treated groups, but still low in the dual treatment group (Supplementary Fig. 6c). By day 5, the ratio of CD8$^+$/Treg

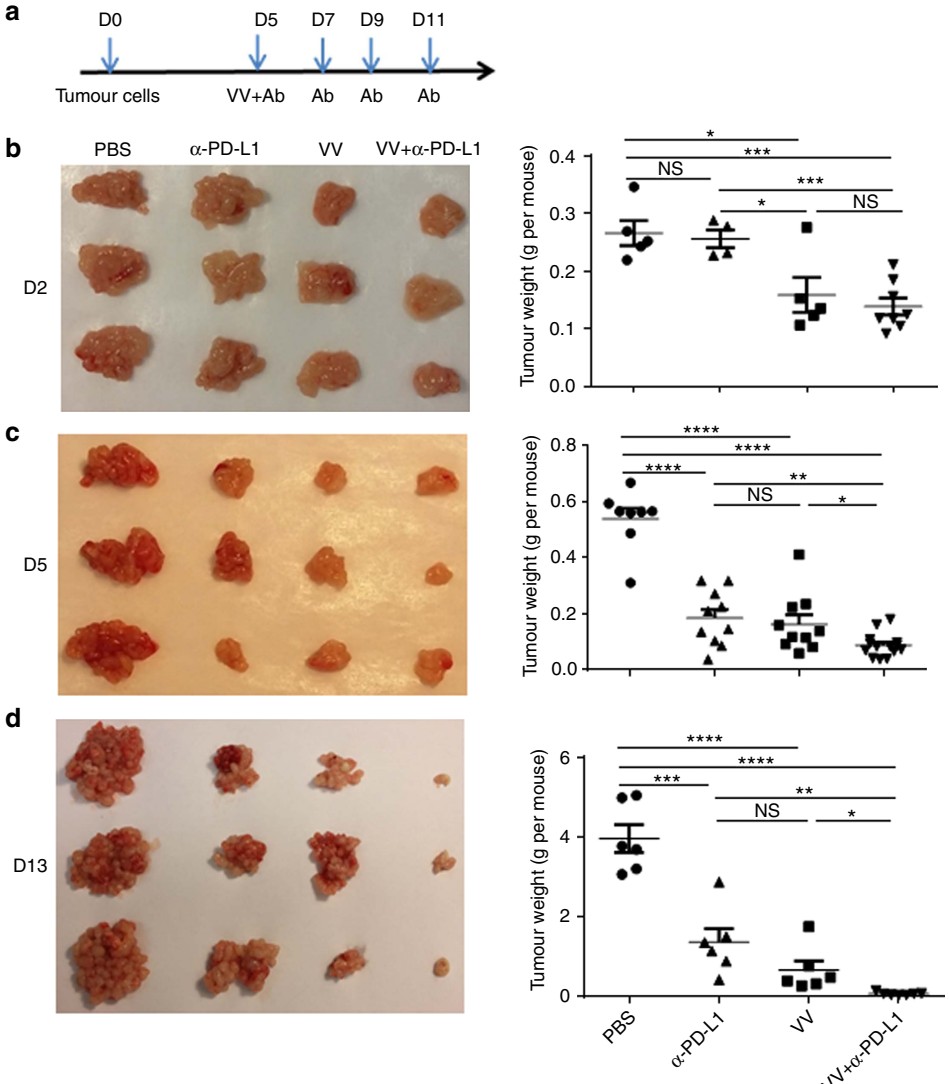

**Figure 3 | Primary tumour growth is dynamically monitored post treatments.** B6 mice were intraperitoneally inoculated with $5 \times 10^5$ MC38-luc cancer cells and treated as described. Tumour-bearing mice were killed on days 2, 5 and 13 after first treatment, and primary tumour tissues were collected. Shown are schematic representations of the experiment (**a**) or weight of primary tumours on day 2 (**b**), day 5 (**c**) and day 13 (**d**) post first treatment. Numbers of mice per group are 5–11 ($n = 5 \sim 11$). In photographs, only representative three tumours from each group are shown. In the figures, VV = vvDD-CXCL11. Data were presented as individuals and means. Student's *t*-test was used to analyse the statistical significance (\*$P < 0.05$; \*\*$P < 0.01$; \*\*\*$P < 0.001$; \*\*\*\*$P < 0.0001$; and NS: not significant).

was quite low in PBS or α-PD-L1 Ab groups, however, increased in the VV -treated group, and further increased in the dual treatment group ($P < 0.01$) (Fig. 6j). The percentage of effector CD4$^+$ T cells (Foxp3$^-$CD4$^+$) in tumour infiltrating CD45$^+$ cells was significantly smaller in the dual treatment group on day 2, but higher than other groups by day 5 (Supplementary Fig. 6d,e; $P = 0.05$).

Increased levels of killer cell activity markers in the TME (IFN-γ, granzyme B and perforin) were also observed in tumours treated with combination therapy. At day 2, IFN-γ was slightly elevated in α-PD-L1 Ab treated tumours, but higher in VV or the dual treatment groups—about fivefold over the PBS group (Supplementary Fig. 7a). By day 5, IFN-γ was about 200-fold higher in the dual treatment group compared to the PBS group, while the increase remained relatively low in the VV or α-PD-L1 Ab treated groups (Fig. 7a). Similar dramatic changes of expression were also observed in both granzyme B and perforin (Fig. 7b,c, Supplementary Fig. 7). Together, these results

demonstrate that dual treatment leads to enhanced infiltration of CD4$^+$ and CD8$^+$ effector T cells, with elevated activation markers (elevated granzyme B, perforin and IFN-γ), as well as reduced T*regs* in the tumour tissues.

**Dual therapy elicited systemic and potent anti-tumour immunity.** To address if the dual treatment could activate the systemic tumour-specific cellular immunity, we isolated splenic CD8$^+$ T cells from the tumour-bearing mice receiving different treatments and re-stimulated with mitomycin C-treated MC38-luc or control B16 cancer cells in the presence of irradiated CD8-depleted splenocytes from naive B6 mice *in vitro* for 48 h. The concentration of IFN-γ was significantly increased in the supernatant of the cultured CD8$^+$ T cells from the dual treatment compared with either monotherapy or PBS (Fig. 8a). This IFN-γ response is quite tumour-specific (Fig. 8a). To investigate whether the dual treatment resulted in the generation of

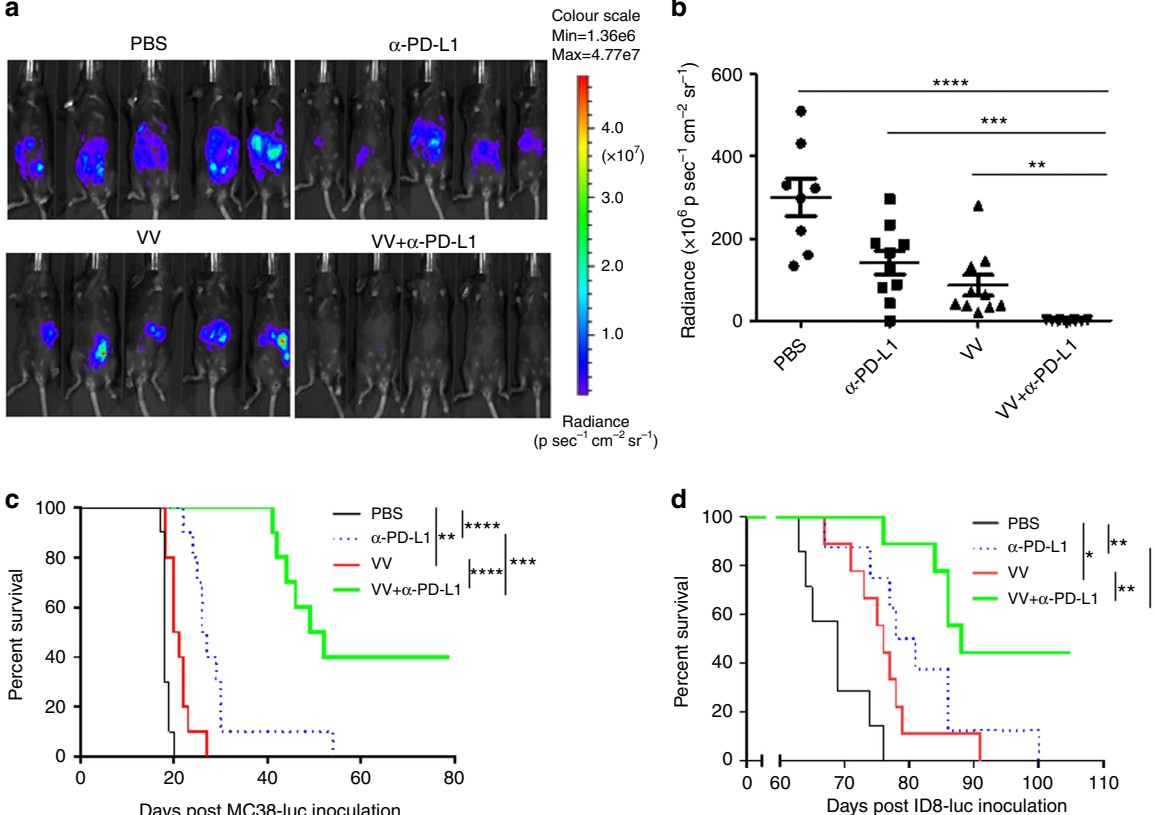

**Figure 4 | VV and PD-L1 blockade synergistically elicit anti-tumour effect.** B6 mice were intraperitoneally inoculated with $5 \times 10^5$ MC38-luc or $3.5 \times 10^6$ ID8-luc cells, respectively, and divided into required groups according to tumour growth condition based on live animal IVIS imaging 5 days post tumour cell injection. Grouped mice were intraperitoneally injected with VV ($2 \times 10^8$ pfu per 100 µl), α-PD-L1 Ab, VV plus α-PD-L1 Ab, or PBS (100 µl) per mouse, respectively. α-PD-L1 Ab (200 µg per 100 µl) was injected every 2 days for a total of four times, according to the scheme in Fig. 3a. Live animal imaging of the mice with MC38-luc tumours at day 9 post first treatment (**a**). The data are quantified ($n = 8 \sim 10$) (**b**). The survival of tumour-bearing mice was monitored by Kaplan–Meier analysis and statistical analyses were performed with Log rank test. Both MC38-luc (**c**) and ID8-luc (**d**) -tumour-bearing mice were shown.

prolonged protective anti-tumour immunity, the dually treated MC38-luc-bearing mice who survived for more than 60 days were re-challenged with MC38-luc s.c. at a higher dose ($1 \times 10^6$ cells) and naive B6 mice received the same treatment as a control. The primary tumour grew much faster on naive mice than that on VV plus α-PD-L1-treated mice (Fig. 8b). These results demonstrate that the dual therapy elicited systemic immunity.

**Both CD4 and CD8 T cells and IFN-γ are needed for therapy.** We then performed a cell depletion experiment to analyse the functions of the infiltrated CD4$^+$, CD8$^+$ T cells and IFN-γ in the therapy. The MC38 colon cancer-bearing mice receiving the dual treatment were injected with depletion Ab at the time points indicated (Fig. 8c). These mice were injected with α-CD8 Ab on days 10, 11, 12, or α-CD4 Ab injection at days 10, 15, 20 post tumour cell inoculations. The mice receiving either anti-CD8 Ab or anti-CD4 Ab died earlier from tumour progression than those receiving the dual treatment without depletion (Fig. 8d). We also studied whether IFN-γ was required for the therapeutic effect. The circulating IFN-γ was neutralized by injection of α-IFN-γ Ab every 2 days beginning on day 10 for a total of four injections and this partially abolished the therapeutic effect (Fig. 8d). In summary, these results demonstrate that CD8$^+$, CD4$^+$ T cells and IFN-γ all play essential roles in the therapeutic efficacy of the dual therapy.

## Discussion

Both oncolytic virotherapy and immune checkpoint blockade have shown great promise as novel classes of drugs to treat certain types of advanced cancers in the last 5 years[8,10]. Yet only a limited percentage of cancer patients achieve objective clinical responses through these novel treatments, leaving significant room for improvement partly due to complicated regulatory circuits of T-cell functions in cancer[31]. In clinical trials, tumour infiltrating T cells and PD-L1 expression[32,33] are both known to indicate the potential for response to α-PD-L1 treatment. By using an oncolytic poxvirus that is designed to attract T cells with high efficiency (CXCL-11 expression) and upregulate PD-L1 (IFN-γ response to poxvirus infection as one of the mechanisms), the transformation of anti-PD-L1 resistant tumours into sensitive tumours is feasible. As an alternative perspective, oncolytic virotherapy is most effective as an immune therapy, yet it induces immunosuppressive factors for its own protection. As a result, the combination of checkpoint inhibitors with OVs should enhance its efficacy.

Accordingly, we hypothesized that the combination of an OV expressing CXCL11 and α-PD-L1 antibody would be an ideal regimen to treat tumours. Investigators have been exploring combination strategies in order to enhance the overall efficacy of these novel treatment strategies while limiting the toxicity. Indeed, a number of studies have explored the combination of an OV with checkpoint blockade in cancer models[34–38], and ongoing clinical trials are addressing this. These studies have

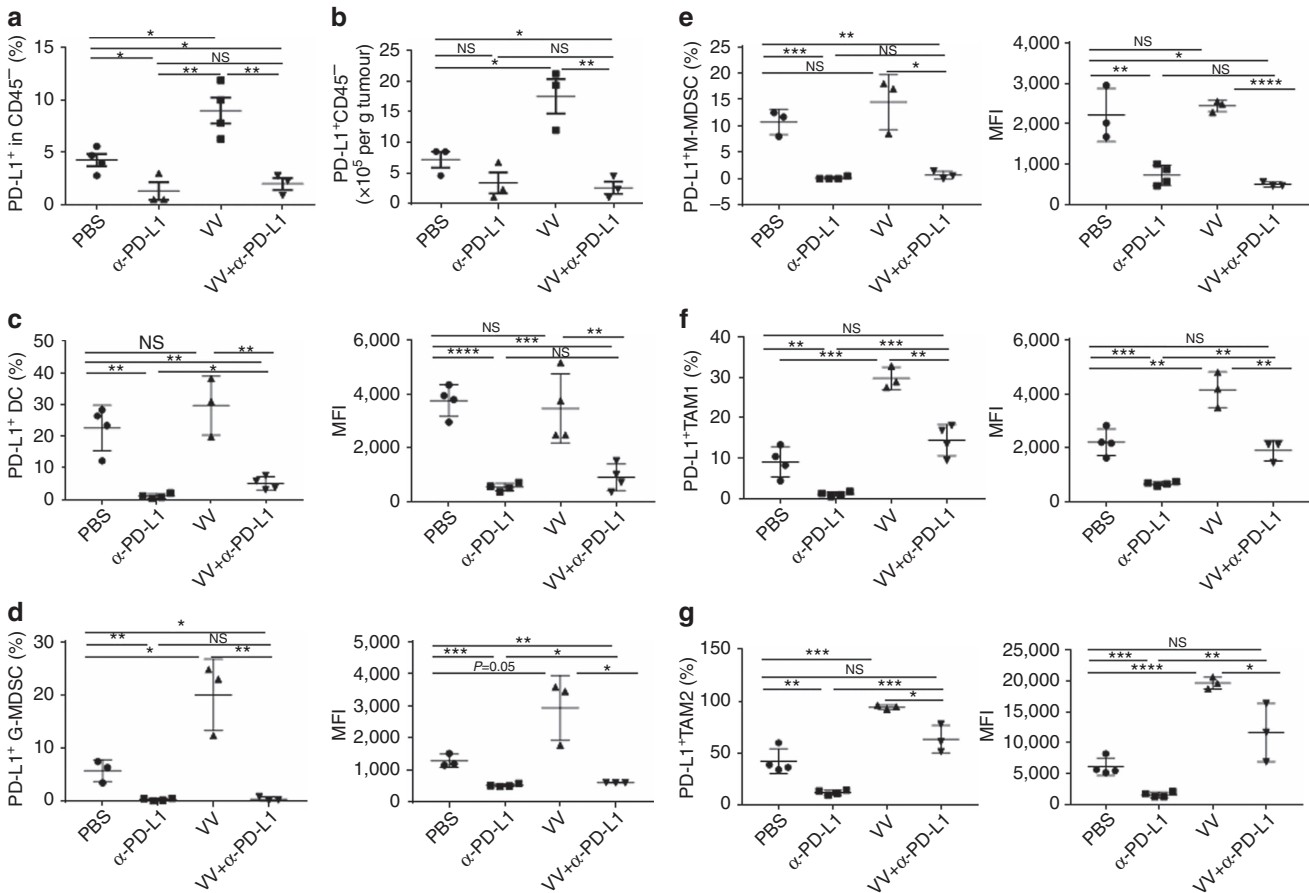

**Figure 5 | The α-PD-L1 treatment reduces the PD-L1$^+$ cells in TME.** B6 mice were intraperitoneally inoculated with $5 \times 10^5$ MC38-luc cancer cells and treated with VV and/or α-PD-L1 as described. Tumour-bearing mice were killed at day 5 post first treatment and primary tumours were collected and analysed to determine the PD-L1$^+$ CD45$^-$ cells (**a,b**), PD-L1$^+$ DC (defined as CD45$^+$ CD11b$^+$CD11c$^+$Ly6G$^-$PD-L1$^+$) (**c**), PD-L1$^+$G-MDSC (defined as CD45$^+$CD11c$^-$CD11b$^+$Ly6G$^+$Ly6c$^{low}$PD-L1$^+$) (**d**), PD-L1$^+$ M-MDSC (defined as CD45$^+$CD11c$^-$CD11b$^+$Ly6G$^-$Ly6c$^{hi}$PD-L1$^+$) (**e**), PD-L1$^+$ TAM1 (defined as CD45$^+$CD11c$^-$CD11b$^+$Ly6G$^-$F4/80$^+$CD206$^-$PD-L1$^+$) (**f**), PD-L1$^+$ TAM2 (defined as CD45$^+$CD11c$^-$CD11b$^+$Ly6G$^-$F4/80$^+$CD206$^+$PD-L1$^+$) (**g**). Of note, the anti-PD-L1 antibody clone 10F.9G2 was used for therapy while clone MHI5 was used for subsequent analysis. Data were analysed using Student's $t$-test ($^*P<0.05$; $^{**}P<0.01$; $^{***}P<0.001$; $^{****}P<0.0001$).

demonstrated that viral infection overcomes tumour resistance to immune checkpoint blockade immunotherapy[36,39]. Radiation has been shown to increase PD-L1 in the TME and therefore can synergize with α-PD-L1 therapy and significantly improve therapeutic efficacy in mice. Our study shows for the first time that an OV induces the expression of PD-L1 on a wide variety of cancer cells and immune cells. The combination of CXCL11-expressing OV plus α-PD-L1 significantly reduced tumour burden and achieved better survival compared to monotherapy. Previous studies have found that PD-L1 expression in the TME is modulated by a number of factors, including HIF-1α (ref. 40), types 1 and 2 interferons, and TNF-α (refs 41,42). These cytokines are released by host cells in response to viral infection, and is likely one of the mechanisms for increased PD-L1 expression on tumour cells.

We believe that this dual treatment works through multiple mechanisms of action. The OV replicates and kills infected cancer and stromal cells, releasing potent danger signals (DAMPs and PAMPs), as well as tumour-associated antigens for dendritic cells to trigger anti-tumour immunity[1,43]. At the same time, virus infection induces a proinflammatory response leading to enhanced expression of cytokines and chemokines, including TNF-α and types I and II TNFs (especially IFN-γ), which induce the expression of PD-L1 and related proteins[41,42].

In addition, OVs such as oncolytic vaccinia virus disrupt tumour-associated vasculature[44], and thus increase further the hypoxic TME, likely leading to elevated PD-L1 expression in cancer cells and infiltrated MDSCs and TAMs (ref. 40). The expression of PD-L1 on cancer and immune cells sensitize the cells as highly susceptible targets for anti-PD-L1 antibody-mediated therapy. Therefore, the PD-L1 antibody targets not only a number of types of immune cells including MDSCs and TAMs to relieve the immunosuppression functions, but also cancer cells that have survived the OV attack yet express PD-L1, which could promote tumour growth (even independent of an immune mechanism)[45]. This synergistic action of the two novel classes of anticancer agents, would lead to a broader indication across a larger patient population and multiple histologies.

Analysis of the cellular components in the TME showed that VV treatment alone recruited effector T cells (especially CD8$^+$ T cells), and α-PD-L1 treatment alone lowered the PD-L1$^+$ cells in the TME, while the combination lowered the PD-L1$^+$ cells and elevated tumour-infiltrating effector T cells in the TME, leading to a significant anti-tumour response. Vaccinia virus elicited a host antiviral immune response, but also led to the recruitment of immune suppressor cells. T-cell-suppressive PD-L1$^+$ MDSC or PD-L1$^+$ TAM were elevated in the virus-treated TME. Previously, MDSC, especially G-MDSC,

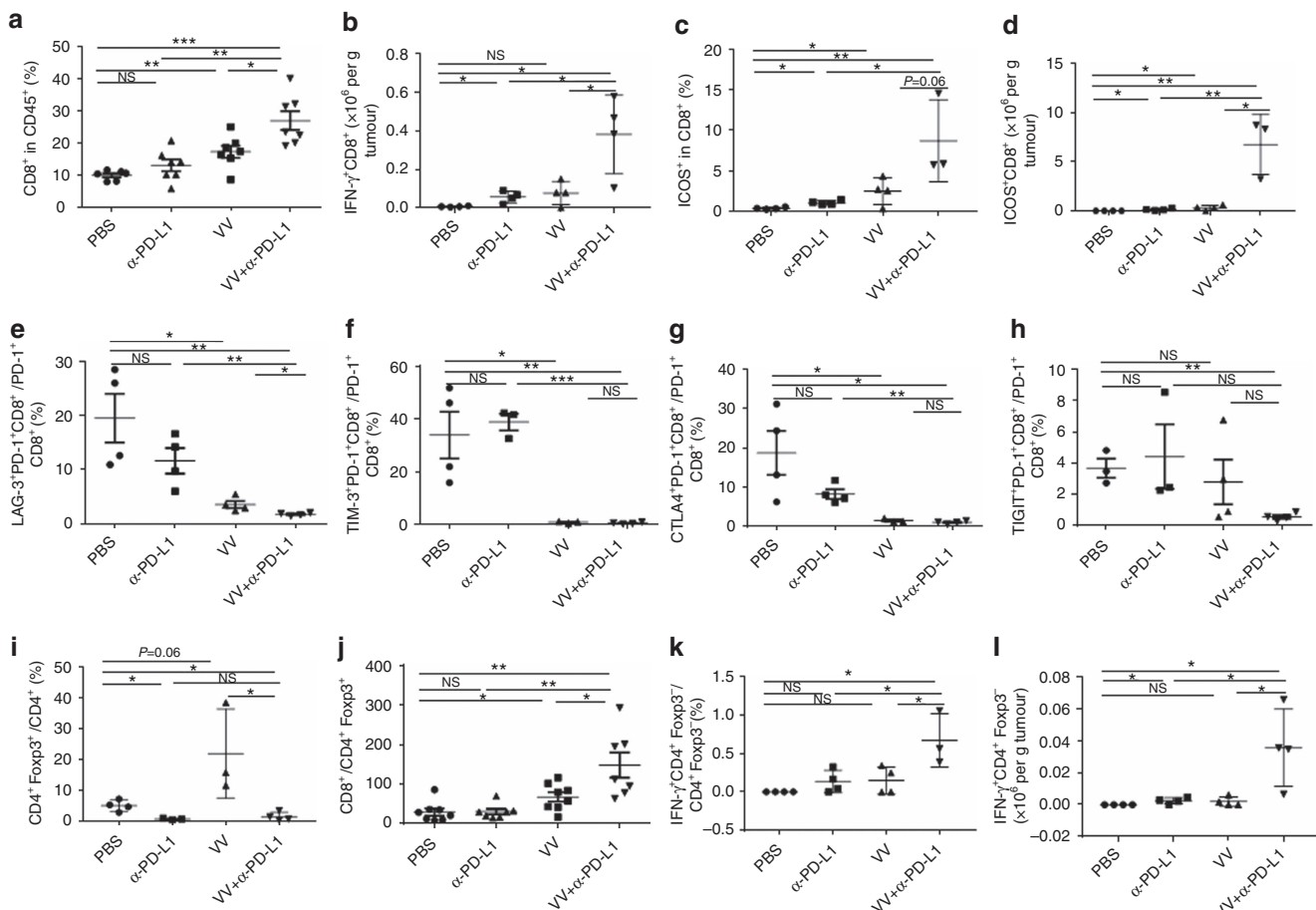

**Figure 6 | The combination of VV plus α-PD-L1 treatment enhances infiltration of effector T cells and reduces T$reg$ cells and exhausted CD8$^+$ T cells in the TME.** B6 mice were intraperitoneally inoculated with $5 \times 10^5$ MC38-luc and treated with VV and/or α-PD-L1 as described. Single cells were made from primary tumours collected from tumour-bearing mice at day 5 post first treatment, blocked with α-CD16/32 Ab and then stained with antibodies against CD45, CD8, CD4, PD-1, ICOS, PD-1, CTLA-4, TIM-3, LAG-3, TIGIT and Foxp3 to determine the quantities of CD8$^+$ T cells (**a**), CD8$^+$ T-cell activation (**b–d**), CD8$^+$ T-cell exhaustion (**e–h**), Treg cells (**i**), CD8$^+$/CD4$^+$Foxp3$^+$ T cells (**j**) and CD4$^+$Foxp3$^-$ T cells (**k,l**) in the TME. Of note, the anti-PD-L1 antibody clone 10F.9G2 was used for therapy while clone MHI5 was used for subsequent phenotypic analysis. Data were analysed using Student's $t$-test (*$P<0.05$; **$P<0.01$; ***$P<0.001$).

was found to be rapidly recruited to the site of vaccinia virus infection for the suppression of virus clearance by NK cells[29]. In the current study, the dual treatment effectively reduced the number and intensity of PD-L1 expression in the subclasses of MDSC and TAM cells (G-MDSC, M-MDSC, TAM1 and TAM2) in the TME, suggesting that it reversed the suppression of tumour-specific T-cell activation elicited by PD-L1$^+$ MDSC or PD-L1$^+$ TAM cells[40,46]. In addition, T$reg$ was reduced too. More surprisingly, the dual treatment greatly diminished exhausted CD8$^+$ T cells by reducing the co-inhibitory molecules double-positive CD8$^+$ T cells (LAG3$^+$PD-1$^+$; TIM-3$^+$PD-1$^+$; CTLA4$^+$PD-1$^+$; TIGIT$^+$PD-1$^+$), which have been shown to be more severely exhausted CD8$^+$ T cells. In summary, through diminishing inhibitory MDSC, TAM, Treg, co-inhibitory molecules-double positive CD8$^+$ T cells, and increasing effector CD4$^+$ T cells, CD8$^+$ T cells (including IFN-γ$^+$, ICOS$^+$, high levels of perforin and granzyme B), the dual treatment enhanced the potent anti-tumor immunity in the TME, thus displayed potent anti-tumour activity and prolonged the survival of tumour-bearing mice.

The timing of the combined therapy is critical, as the virus has a marked direct cytotoxic effect from replicative oncolysis; then as a result of immunogenic cancer cell death

and immune stimulating transgenes, an anti-tumour immune response is generated. It is important to achieve both effects for maximum results. If the virus is cleared by the immune system before spreading throughout the tumour, the overall anti-tumour effect is diminished. On the other hand, replicating virus induces PD-L1 in the TME which may impact the development of an effective anti-tumour immune response. If given first, α-PD-L1 therapy may lead to premature clearance of the virus, and a diminution of anti-tumour activity. If α-PD-L1 Ab is given too late, the upregulated PD-L1 level is too high to be effectively blocked by the anti-PD-L1 antibody, and the virus will have been immunologically cleared before anti-tumour immunity can be generated. Our data support the best timing to be simultaneous delivery of both agents. This supports the idea that an anti-PD-L1 approach could be incorporated into the virus without the need for delayed expression of the transgene.

The OV and α-PD-L1 antibody worked together at the tumour site and subsequently induced systemic anti-tumour immunity. The combination worked best to increase tumour-specific IFN-γ-producing CD8$^+$ T cells in the spleen, and re-challenge of cured mice after combination therapy resulted in marked attenuation of tumour growth. We found that VV treatment led to elevated PD-L1 expression not only in cancer cells, but

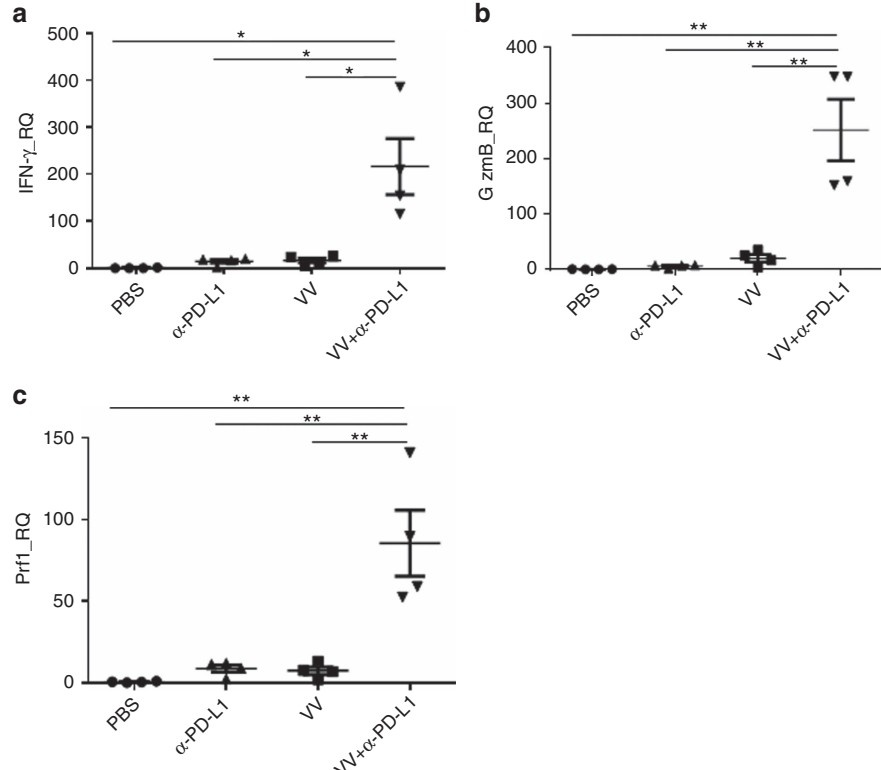

**Figure 7 | The combination therapy leads to enhanced activation markers in the TME.** In one experiment, the i.p. MC38-luc tumour model was treated with VV and/or α-PD-L1 as described. Tumour tissues were collected at day 5 post first treatment and were used for extraction of total RNA. RT–qPCR assays were performed to determine the levels of IFN-γ (**a**), granzyme B (**b**) and perforin (**c**) in the TME. Data were analysed using Student's *t*-test ($^*P < 0.05$; $^{**}P < 0.01$).

also in immune cells. Higher PD-L1 in the TME may be an immune escape mechanism for the virus, and this may suppress systemic anti-tumour immunity. It has previously been demonstrated that PD-L1$^+$ MDSC or PD-L1$^+$ TAM could effectively suppress tumour-specific immunity[40,46]. PD-L1$^+$ MDSC and PD-L1$^+$ TAM were both increased in the TME post virus treatment in the current study, and may have an impact on systemic anti-tumour immunity. In the dual therapy setting, α-PD-L1 Ab therapy reversed the increase of these immunosuppressive cell types, and greatly reduced the FoxP3$^+$CD4$^+$ T*reg* cells and exhausted CD8$^+$ T cells, leading to improved systemic immunity. It is possible that this systemic response can lead to an abscopal effect in tumours not infected with the virus, but this was not tested here. In a previous study, the authors reported that oncolytic virotherapy overcomes systemic tumour resistance to immune checkpoint blockade, due to the virus-induced inflammatory response which coincided with distant tumour infiltration of tumour-specific CD4$^+$ and CD8$^+$ T cells[36]. In another study, the authors concluded that PD-1 blockade did not affect the magnitude of oncolysis-mediated anti-tumour responses, but rather broadened the spectrum of T-cell responses by elucidating more neoepitopes[39]. As for therapeutic setting, we have noticed that either monotherapy worked well in a tumour model when the tumour burden was minimal. Thus, the adjuvant stetting is an ideal setting for this approach. In the current study, we have not studied in detail the toxicity associated with the dual therapy, even though no severe toxicity was observed under these conditions.

In summary, we have demonstrated that an oncolytic vaccinia virus markedly upregulates PD-L1 in the TME, and thereby synergizes with α-PD-L1 treatment leading to over 40% cures in aggressive models of peritoneal carcinomatosis from colon and ovarian cancers. Multiple histologies tested demonstrated similar upregulation of PD-L1 in response to vaccinia virus infection. The action of OV results in an increased number of cancer types sensitized to anti-PD-L1 antibody therapy. These two classes of novel anticancer agents work synergistically to exert cytotoxicity to cancer cells, eliminate immunosuppressive cells (including MDSC, TAM, T*reg* and exhausted CD8$^+$ T cells), and elicit more potent and sustained systemic anti-tumour immunity, thus achieving better therapeutic efficacy. Therefore, our study provides a rational strategy of combination therapy consisting of VV and α-PD-L1 antibody for a broad range of cancer patients.

## Methods

**Mice and cell lines.** Female C57BL/6 (B6 in short) mice were purchased from The Jackson Laboratory (Bar Harbor, ME, USA) and housed in specific pathogen-free conditions in the University of Pittsburgh animal facility. All animal studies were approved by the Institutional Animal Care and Use Committee of the University. ID8-luc, a B6 mouse-derived ovarian cancer cell line tagged with firefly luciferase gene, was kindly provided by Dr Natasa Obermajer (University of Pittsburgh). Mouse colon cancer MC38-luc, mesothelioma AB12-luc and MOSEC cancer cell lines were generated as described previously[47]. Murine mammary carcinoma EMT-6 was used in our previous study[48]. Murine melanoma B16, fibrosarcoma MCA102, hepatoma Hepa1-6, human cancer cell lines (HCT116, LS174T, HT-29, DLD1, HepG2, H1299, MDA468, HaCAT, REN, Hela and A2780) were obtained from ATCC (Manassas, VA, USA). All cell lines were grown in Dulbecco's modified Eagle's medium supplemented with 10% fetal bovine serum (FBS), L-glutamine and penicillin/streptomycin (Invitrogen, Carlsbad, CA, USA) in 37 °C, 5% $CO_2$ incubator.

**Viruses and antibodies.** Recombinant Vaccinia virus (Western Reserve strain) vvDD-DsRed (vvDD) and vvDD-CXCL11 (VV) were previously described[14]. Anti-mouse PD-L1 Ab (clone 10F.9G2), α-mouse CD8 Ab (clone 53-6.7), α-mouse

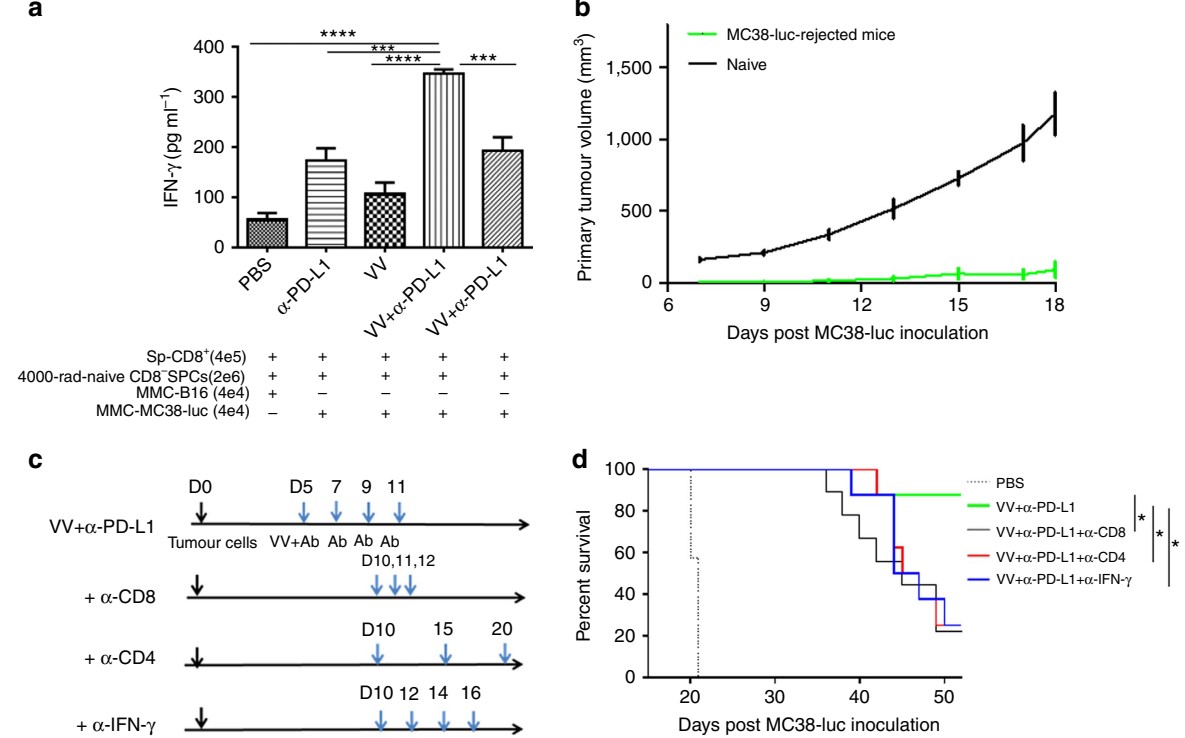

**Figure 8 | The systemic anti-tumour immunity elicited by dual therapy plays an important role in the overall therapeutic efficacy. (a)** B6 mice were intraperitoneally inoculated with $5 \times 10^5$ MC38-luc cancer cells and treated with VV and/or α-PD-L1 as described. Splenic CD8$^+$ T cells ($4 \times 10^5$) were isolated from naive and MC38-luc-bearing mice that received different treatments 18 days post tumour cell injection and restimulated with mitomycin C-treated MC38-luc or B16 cancer cells ($4 \times 10^4$ cells each) in the presence of 4000-rad-irradiated CD8-depleted naive B6 splenocytes ($2 \times 10^6$) in 200 μl RPMI-1640 medium supplemented with 10% FBS at 37 °C, 5% $CO_2$ for 2 days. The concentration of IFN-γ in the culture supernatants was tested by ELISA. The statistical analyses were performed with $t$-test. **(b)** Naive or MC38-luc-bearing B6 mice with dual treatments, which survived for more than 60 days, were s.c. rechallenged with $1 \times 10^6$ MC38-luc cancer cells. The primary tumour size was measured and presented here. **(c)** In a separate experiment, B6 mice were inoculated with $5 \times 10^5$ MC38-luc cells i.p. and treated with VV plus α-PD-L1 or PBS at day 5 post tumour inoculation, α-PD-L1 Ab was injected every 2 days for a total of four times. α-CD8 Ab (250 μg per injection), α-CD4 Ab (150 μg per injection) or α-IFN-γ Ab (200 μg per injection) were intraperitoneally injected into mice to deplete CD8 + T cells, CD4 + T cells or neutralize circulating IFN-γ as scheduled in **c**, and the overall survival was monitored by Kaplan–Meier analysis and analysed using log rank test (**d**).

CD4 Ab (clone GK1.5) and α-mouse IFN-γ Ab (clone XMG1.2) were purchased from Bio X Cell (West Lebanon, NH, USA).

**Viral infection of cancer cells *in vitro*.** MC38-luc ($4 \times 10^5$) or ID8-luc ($4 \times 10^5$) cells were seeded in six-well plates overnight and then infected with vvDD at MOI of 1.0 in 0.6 ml of 2% FBS-containing Dulbecco's modified Eagle's medium for 2 h. Growth medium (CM) (1.4 ml) was added for culture and cells were collected around 24 h post virus infection.

**Rodent tumour models.** For subcutaneous (s.c.) tumour model, B6 mice were subcutaneously inoculated with $5 \times 10^5$ MC38-luc cancer cells. PBS or vvDD was intratumorally injected at $2 \times 10^8$ pfu per tumour when the s.c tumour area reached $5 \times 5 \, mm^2$. Tumour tissues were collected from PBS or virus-treated mice 4 days post treatment.

For peritoneal (i.p.) tumour models, B6 mice were intraperitoneally inoculated with $5 \times 10^5$ MC38-luc or $3.5 \times 10^6$ ID8-luc cancer cells, respectively, and divided into required groups according to tumour size based on live animal IVIS imaging 5 days post tumour cell injection, performed using a Xenogen IVIS 200 Optical *In Vivo* Imaging System (Caliper Life Sciences, Hopkinton, MA, USA). Grouped mice were intraperitoneally injected with VV ($2 \times 10^8$ pfu per 100 μl), α-PD-L1 Ab (clone 10F.9G2, 200 μg per injection), VV plus α-PD-L1 Ab, or PBS (100 μl) per mouse, respectively. Anti-PD-L1 Ab was injected once every 2 days for a total of four times. In indicated groups, VV or α-PD-L1 Ab were injected at different time points. In some experiments, α-CD8 Ab at 250 μg per injection (clone 53-6.7; Bio X Cell), α-CD4 Ab (clone GK1.5, Bio X Cell; 150 μg per injection), or α-IFN-γ Ab (clone XMG1.2, Bio X Cell; 200 μg per injection) were intraperitoneally injected into mice to deplete CD8 + T cells, CD4 + T cells or neutralize circulating IFN-γ. In some experiments, mice were killed to collect tumour tissues and spleens at indicated time points.

MC38-luc-tumour-bearing B6 mice treated with VV and α-PD-L1 Ab, which survived more than 60 days, were re-challenged s.c. with $1 \times 10^6$ MC38-luc cells

per mouse. And naive B6 mice received same dose tumour challenge as a control. The primary tumour size was measured using an electric calliper in two perpendicular diameters.

**Flow cytometry.** Collected tumour tissues were weighed, minced and incubated in RPMI 1640 medium containing 2% FBS, 1 mg ml$^{-1}$ collagenase IV (Sigma: #C5138), 0.1 mg hyaluronidase (Sigma: #D5025) and 200U DNase I (Sigma: H6254) at 37 °C for 1-2 h to make single cells. *In vitro* virus-infected cells or single cells from tumour tissues were blocked with α-CD16/32 Ab (clone 93, eBioscience: #14-0161-85) and then stained with antibodies against mouse CD45 (Alexa 700 or PerCP-Cy5.5, clone: 30-F11, BioLegend: #103128 or #103132, 1:300), CD11b (PE, clone: M1/70, BioLegend: #101208, 1:300), Gr1 (FITC, clone: RB6-8C5, BioLegend: #108406, 1:300), PD-L1 (APC, clone: 10F.9G2, BioLegend: #124312, 1:300; Biotin, clone: MHI5, eBioscience: #13-5982-85, 1:300 + Brilliant Violet 421-Streptavidin, BioLegend: #405226, 1:1000), Ly6G (APC, clone: 1A8, BioLegend: #127614, 1:300), Ly6c (PerCP-Cy5.5, clone: HK1.4, BioLegend: #128012, 1:300), CD11c (PE-CF594, clone: HL3, BD Biosciences: #562454, 1:300), F4/80 (FITC or PE-Cy7, clone: BM8, eBioscience: #12-4801-82 or 25-4801-82, 1:300), CD206 (FITC, clone: MR5D3, Bio-Rad, #MCA2235F, 1:100), CD4 (FITC or PE-CF594, clone: RM4-5, BD Biosciences: #561835 or 562285, 1:300), CD8 (PEor APC/Cy7, clone:53-6.7, BioLegend: #100708 or 100714, 1:300), CTLA-4 (PerCP-Cy5.5, clone: UC10-4B9, BioLegend: #106316, 1:300), TIM-3 (clone:8B.2C12, Biotin, eBioscience: 13-5871-82, 1:300 + APC-Streptavidin, eBioscience: #17-4317-82, 1:1,000), LAG-3 (eFluor 450, clone: C9B7W, eBioscience: #48-2231-82, 1:300), TIGIT (PE-Cy7, clone: 1G9, BioLegend: #142108, 1:300), ICOS (PE, clone: 7E.17G9, BioLegend: #117406, 1:300), NK1.1 (PE-Cy7, clone: PK136, BioLegend: #108714, 1:300), or intracellular stained with Foxp3 (APC or PE, clone: FJK-16s, eBioscience: #17-5773-82 or 12-5573-82, 1:100) and IFN-γ (APC, clone: XMG1.2, e-Bioscience: #17-7311-82, 1:100) following the instruction of FOXP3 Fix/Perm Buffer Set (BioLegend). Samples were collected on BD Accuri C6 cytometer or Beckman Coulter Gallios, and data were analysed using BD Accuri C6 cytometer software and FlowJo software (Tree Star Inc., Ashland, OR).

**RT–qPCR.** Total RNA was extracted from virus-infected cells or tumour tissues using the RNeasy Kit (Qiagen, Valencia, CA, USA). One microgram of RNA was used for cDNA synthesis, and 25–50 ng of subsequent cDNA was used to conduct mRNA expression analysis by TaqMan analysis on the StepOnePlus system (Life Technologies, Grand Island, NY, USA). All the primers for the analysis were purchased from Thermo Fisher Scientific (Waltham, MA, USA). The gene expression was normalized to a house-keeping gene HPRT1 and expressed as fold increase $(2^{-\Delta CT})$, where $\Delta CT = CT_{(Target\ gene)} - CT_{(HPRT1)}$.

**Systemic anti-tumour immune response.** Splenic $CD8^+$ T cells were isolated from naive and MC38-luc-bearing B6 mice received different treatment 18 days post tumour cell injection using α-mouse CD8 microbeads following vendor's protocols (Miltenyi Biotec, San Diego, CA, USA). These cells ($4 \times 10^5$ cells per assay) were restimulated with mitomycin C-treated MC38-luc cells or B16 cells (tumour-specific control) in the presence of 4,000-rad-irradiated CD8-depleted splenocytes from naive B6 mouse ($2 \times 10^6$) in 200 µl RPMI 1640 medium supplemented with 10% FBS at 37 °C, 5% $CO_2$ for 2 days. The concentration of IFN-γ in the culture supernatants was tested using ELISA kit according to the manufacturer's instructions (BioLegend).

**Statistics.** The data presented in the figures are mean ± s.d. Statistical analyses were performed using Student's t-test (GraphPad Prism version 5). Animal survival is presented using Kaplan–Meier survival curves and was statistically analysed using log rank test (GraphPad Prism version 5). Value of $P < 0.05$ is considered to be statistically significant, and all P values were two-sided. In the figures, the standard symbols were used: $*P < 0.05$; $**P < 0.01$; $***P < 0.001$ and $****P < 0.0001$.

**Data availability.** All relevant data are available from the authors on request.

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

## Acknowledgements

This work has been supported by NIH grants P01CA132714 and R01CA155925. This project used the University of Pittsburgh Cancer Institute shared facilities that are supported in part by the NIH grant award P30CA047904. This work was also partially funded by generous support for mesothelioma research from the New Era Cap Company and from Valarie Koch.

## Author contributions

Z.L., R.R., Z.S.G. performed the experiments. Z.L., Z.S.G. and D.L.B. designed the experiments and analysed the data. Z.L., Z.S.G. and D.L.B. wrote the paper. P.K. provided advice and made suggestions for the manuscript.

## Additional information

**Competing interests:** D.L.B. is a shareholder of Sillajen Biotheraputics, a company developing oncolytic viruses. The remaining authors declare no competing financial interests.

