## [Peer Review File · Nature Communications]

Reviewers' comments:

Reviewer #1 (Remarks to the Author):

In the manuscript "Rational Combination of CXCL11-Expressing Oncolytic Virus and PD-L1 Blockade Works Synergistically to Enhance Therapeutic Efficacy" Liu et al. described a rationale for a new combination therapy with a CXCL-11 expressing oncolytic virus (VV) and anti PD-L1 antibody. The authors demonstrated an increase of PD-L1 expression after VV treatment in colon and ovarian cancer models. They also showed an increase in CD8+ T cell infiltration and cytokine expression in tumor after the combination therapy.

This is a well written and fairly good manuscript, with interesting results that can be promising for future clinical application. However, the manuscript would gain in significance if an orthotopic model is used (Tseng et al. Jove 2007), and the overall quality of the data has to be substantially improved.

Major points

1. The authors show an increase in PD-L1 expression by real-time PCR and flow cytometry after VV treatment. Although, in Figure 1a the real-time PCR shows significant ($p < 0.01$) increase of PD-L1 expression, the fold-change difference is very small. Please explain this result.
2. In the paper Liu et al. consider CD45- as tumor cells. This is not correct, indeed multiple cells can be CD45-, i.e. endothelial cells, hematopoietic cells, fibroblasts...
3. The markers used to discriminate MDSCs and TAMs are not enough to describe these populations of cells. In particular, Gr1 and F4/80 are not the best markers for flow cytometry. The authors should consider adding Ly6G and Ly6C to discriminate between monocytes, neutrophils and macrophages. In addition, the paper could gain in impact if DC and NK cells were also analyzed.
4. Flow cytometry analysis shows a down-regulation of PD-L1 (Figure 5) in the in vivo experiments. However, the same PD-L1 clone has been used for treatments and flow cytometry staining. This could potentially result in artefacts.
5. Despite the fact that the authors have evaluated CD8 and CD4/Treg infiltration, the activation of CD8+ T cells has not been analyzed. Moreover, given that authors are suggesting a combination therapy with checkpoint inhibitors, a deeper analysis of T cell activation/exhaustion status should be included.

Minor points

1. Figure 1a - lack of axis labeling.
2. Figure 1b and c: specify whether it is a ratio between infected/mock or what else?
3. Add axis labeling to figure 3b and 3c.
4. "due to the virus-induced direct oncolysis"(page 7): in this case the authors are speculating, this sentence should be revised.
5. Which model did you used in figure 4a?
6. Figure 4 labels (a,b,c,d) are mixed. Please adjust that.

Reviewer #2 (Remarks to the Author):

This paper by Lin et al. represents an important contribution in the literature to define use of oncolytic viruses with the use of check point inhibitors in the treatment of cancer. The authors demonstrated: 1) up-regulation of t-regs with oncolytic viral therapy, 2) neutralization of this immunosuppressive activity with anti-PDL1 therapy, 3) synergism of oncolytic viral therapy and check-point inhibition on cancer killing, and 4) importance of the timing for combination therapy.

This paper is worthy of publication. Following are my queries and suggestions for improvement.

1. Is it the presence of viral antigens, replication, lysis or the combinations that lead to increase in t-regs in the area? Have the authors looked at inactivated viruses in their models?
2. What are the toxicities associated with administration of anti-PDL1 therapy? What are the toxicities locally at tumor site and distantly (colon, skin etc.)?
3. Have the authors attempted the treatments even earlier after tumor implantation, and does that produce cures in the animals? This would have implications as to use of such strategies in an adjuvant setting.
4. On figure four, are the survival curves for anti-PDL1 alone statistically significant compared to PBS?
5. What are the comparative results using a vaccinia virus without the chemokine CXCR11? Is the combination of a purely oncolytic virus and checkpoint inhibition effective?

NCOMMS-16-17360

Specific Point-by-Points Responses to the Reviewers by the Authors:

We thank both reviewers for very constructive comments which have led to a much improved manuscript. We have addressed the comments and suggestions by the reviewers in great detail. In the following, reviewers' original comments are in **black** and authors' responses are in **blue**.

Reviewer #1 (Remarks to the Author):

General comments: In the manuscript "Rational Combination of CXCL11-Expressing Oncolytic Virus and PD-L1 Blockade Works Synergistically to Enhance Therapeutic Efficacy" Liu et al. described a rationale for a new combination therapy with a CXCL-11 expressing oncolytic virus (VV) and anti PD-L1 antibody. The authors demonstrated an increase of PD-L1 expression after VV treatment in colon and ovarian cancer models. They also showed an increase in CD8+ T cell infiltration and cytokine expression in tumor after the combination therapy.

This is a well written and fairly good manuscript, with interesting results that can be promising for future clinical application. However, the manuscript would gain in significance if an orthotopic model is used (Tseng et al. Jove 2007), and the overall quality of the data has to be substantially improved.

We appreciate your positive comments.

Your suggestion of using orthotopic model is a very good one. Our clinical practice is in the treatment of patients with peritoneal metastases from colon, ovarian, and mesothelioma. For our translational program, these models mimic our patients and are therefore relevant. In the future, it will be important to examine the impact these treatments can have on the primary tumors, using a true orthotopic model.

Major points:

1. The authors show an increase in PD-L1 expression by real-time PCR and flow cytometry after VV treatment. Although, in Figure 1a the real-time PCR shows significant ($p < 0.01$) increase of PD-L1 expression, the fold-change difference is very small. Please explain this result. Given the virus replication at the wound site, should the perioperative period be an exclusion criteria for injection on future trials? The authors should discuss why or why not.

We have observed the induction of PD-L1, ranging from 2 to 16-fold, in human and murine cancer cells in vitro. It is important to point out that the fold of induction is sometimes misleading. For example, when the basal level of PD-L1 expression is high (such as MC38, at 22%), it is hard to induce to a few more fold (in this case, only to 35.4% cells). When the basal level is low (such as ID8 ovarian cancer cells, at 2.4%), it is easier to induce with high fold (at ~12-fold, but the absolute number of cells stand at 18% only). As a result, PD-L1 is positive in 35.4% MC38 cells and 18% ID8 cancer cells after VV treatment, even though the fold of induction was much higher in ID8 cancer cells than in MC38 cancer cells.

In the previous studies by other investigators, it has been shown that cytokines such as IFN-gamma can induce the expression of PD-L1 in vivo. Thus, the induction of PD-L1 in vitro and in vivo may be mediated through very different mechanisms. There is room for improvements in the future as two or more mechanisms of induction may work in synergy. Nevertheless, the induction we have observed so far, up to 16-fold, are significant both statistically and biologically.

We agree that the perioperative period should be an exclusion criterion for injection on future clinical trials as the virus could replicate at the wound sites. The recovery of replicating virus in a healing wound suggests that systemic delivery is possible, and that the selective mutations of the vaccinia virus do not differentiate tumor tissue from healing or inflamed tissue. It has been previously noted that active psoriatic skin rashes supported vaccinia replication, and it is therefore a contraindication to smallpox vaccination. Based on clinical studies from us and others, while virus recovery was not associated with any significant toxicity, actively healing wounds and acute inflammatory conditions of the skin or oral mucosa in the perioperative period or pathological conditions should be an exclusion criterion in systemic delivery of WR strain-derived oncolytic virus. (a few sentences are added in the Discussion).

2. In the paper Liu et al. consider CD45- as tumor cells. This is not correct, indeed multiple cells can be CD45-, i.e. endothelial cells, hematopoietic cells, fibroblasts...

You are correct. We have corrected the statement.

3. The markers used to discriminate MDSCs and TAMs are not enough to describe these populations of cells. In particular, Gr1 and F4/80 are not the best markers for flow cytometry. The authors should consider adding Ly6G and Ly6C to discriminate between monocytes, neutrophils and macrophages. In addition, the paper could gain in impact if DC and NK cells were also analyzed.

Thank you for the great suggestion. We have used now more markers to classify the groups of MDSCs and TAMs. New data are presented in Figure 5 and supplementary figures 4-5. We have examined subgroups of MDSCs and TAMs: PD-L1⁺.G-MDSC (defined as CD45⁺CD11c⁻CD11b⁺Ly6G⁺Ly6c^{low}PD-L1⁺), PD-L1⁺ M-MDSC (defined as CD45⁺CD11c⁻CD11b⁺Ly6G⁻Ly6c^{hi}PD-L1⁺), PD-L1⁺ TAM1 (defined as CD45⁺CD11c⁻CD11b⁺Ly6G⁻F4/80⁺CD206⁻PD-L1⁺), PD-L1⁺ TAM2 (defined as CD45⁺CD11c⁻CD11b⁺Ly6G⁻F4/80⁺CD206⁺PD-L1⁺).

As you have suggested, we have also examined DC and NK cells. The data on DC are presented in Figure 5c, while data on NK cells are presented in Supplementary Figure 5g.

4. Flow cytometry analysis shows a down-regulation of PD-L1 (Figure 5) in the in vivo experiments. However, the same PD-L1 clone has been used for treatments and flow cytometry staining. This could potentially result in artefacts.

Your point is well taken. For all the experiments where anti-PD-L1 antibody was needed for both treatment and later analysis, two different antibodies were used (Figures 5 and 6). In these studies, anti-PD-L1 antibody, clone 10F.9G2 was used for therapy, while clone

MHI 5 was used for analysis. Both clones of Mab were obtained from the company “Bio X Cell”, to the best of our knowledge, they recognize different epitopes of the PD-L1.

5. Despite the fact that the authors have evaluated CD8 and CD4/Treg infiltration, the activation of CD8+ T cells has not been analyzed. Moreover, given that authors are suggesting a combination therapy with checkpoint inhibitors, a deeper analysis of T cell activation/exhaustion status should be included.

Another great suggestion. We have thus conducted additional experiments to answer this. New data are presented in figure 6 and supplementary figure 6.

Minor points:

1. Figure 1a - lack of axis labeling:

We have now added the axis labeling.

2. Figure 1b and c: specify whether it is a ratio between infected/mock or what else?

The ratio is between infected versus mock-infected cancer cells. We have added the statement to the figure legend.

3. Add axis labeling to figure 3b and 3c.

We have revised the figure and the issue has been resolved.

4. “due to the virus-induced direct oncolysis”(page 7): in this case the authors are speculating, this sentence should be revised.

That phrase has been deleted.

5. Which model did you used in figure 4a?

It was MC38-luc colon cancer model. We have clarified it in the figure legend.

6. Figure 4 labels (a,b,c,d) are mixed. Please adjust that.

Thank you for pointing out the errors. We have now corrected that.

Reviewer #2 (Remarks to the Author):

General comments

This paper by Liu et al. represents an important contribution in the literature to define use of oncolytic viruses with the use of check point inhibitors in the treatment of cancer. The authors demonstrated: 1) up-regulation of t-regs with oncolytic viral therapy, 2) neutralization of this immunosuppressive activity with anti-PDL1 therapy, 3) synergism of oncolytic viral therapy and check-point inhibition on cancer killing, and 4) importance of the timing for combination therapy. This paper is worthy of publication.

Thank you very much.

Following are my queries and suggestions for improvement.

1. Is it the presence of viral antigens, replication, lysis or the combinations that lead to increase in T-regs in the area? Have the authors looked at inactivated viruses in their models?

We did observe some enhanced Treg when the virus was used alone, but not with anti-PD-L1 antibody alone or the combination treatment (Fig. 6i).

We have performed studies with inactivated virus in the past, which had no effect on tumor growth. While we have not included that control in these experiments, it is our impression that replication and both early and late gene expression is required for the complete effect of the virus. In the future, it would be interesting to study what specifically affects the T-reg infiltration.

2. What are the toxicities associated with administration of anti-PDL1 therapy? What are the toxicities locally at tumor site and distantly (colon, skin etc.)?

Toxicities associated with anti-PD-1 and anti-PD-L1 antibodies have been documented in human clinical trials and are a result of autoimmune activity. In our current study with mice, we have not observed any severe toxicities or obvious toxicities under the conditions used. Nevertheless, it is an important issue and we will look into that in details in the combination setting in our future studies.

We have added a few sentences to discuss this issue on the Discussion (page 17).

3. Have the authors attempted the treatments even earlier after tumor implantation, and does that produce cures in the animals? This would have implications as to use of such strategies in an adjuvant setting.

In an earlier tumor model (MC38 colon), our OVs alone have achieved great therapeutic results, with complete tumor-regression and long term survival in a large fraction of mice. Thus, we have not tested this combination strategy in an earlier tumor model. We do feel that an adjuvant setting is ideal for this approach. A simple statement has been added to the Discussion (Page 17).

4. On figure four, are the survival curves for anti-PDL1 alone statistically significant compared to PBS?

Yes it is. In both MC38-luc colon and ID8-luc ovarian tumor models, $p < 0.01$ between PBS and anti-PD-L1 Ab. This is clarified in the Figure and legend.

5. What are the comparative results using a vaccinia virus without the chemokine CXCR11? Is the combination of a purely oncolytic virus and checkpoint inhibition effective?

We have done the experiment with MC38-luc tumor model. We have observed a trend of better results when the CXCL11-armed virus was combined with anti-PD-L1 antibody, as we noticed that three mice survived for at least 120 days in the group with CXCL11-expressing virus while in the other group, all of the mice were dead by day 72. The median survival was also extended from 42.5 days to 47 days. Yet the p value was not significant between the two groups. The data are as follows.

For this reason, we have modified the title and discussion a little bit to de-emphasize the role of CXCL11. We have added the following sentence to the Results section in the manuscript (on page 7):

“It should be noted, however, that direct comparisons between wvDD + α -PD-L1 Ab and VV + α -PD-L1 Ab did demonstrate only a trend of better, yet not statistically significant difference in therapeutic efficacy for the CXCL11 virus (data not shown).”

REVIEWERS' COMMENTS:

Reviewer #1 (Remarks to the Author):

authors have answered all our questions

Reviewer #2 (Remarks to the Author):

The Authors have addressed the reviews adequately. This reviewer is still not convinced by the presented data of the importance of CXCL11 as a chemokine in this model. Given that they have deemphasized this in the manuscript and title, their revisions are acceptable.

NCOMMS-16-17360A

Specific Point-by-Points Responses to the Reviewers by the Authors:

We thank both reviewers for very constructive comments in the first review which have led to a much improved manuscript.

For the revised version of the manuscript, they did not raise any additional questions.